Comparing enzyme activity modifier equations through the development of global data fitting templates in Excel

Walsh Ryan ryan.walsh@iaf.inrs.ca
Microbiology/Biochemistry, INRS–Institut Armand-Frappier , Laval , Quebec , Canada
Tulkens Paul
Electronic publication date: 2018 Dec 14
Publication date: 2018
Volume: 6
Electronic Location ID: e6082
Received 2017 Jul 18; Accepted 2018 Nov 7
Copyright: ©2018 Walsh
Copyright year: 2018
Copyright holder: Walsh
License: This is an open access article distributed under the terms of the Creative Commons Attribution License, which permits unrestricted use, distribution, reproduction and adaptation in any medium and for any purpose provided that it is properly attributed. For attribution, the original author(s), title, publication source (PeerJ) and either DOI or URL of the article must be cited.
License URL: https://creativecommons.org/licenses/by/4.0/

Keywords: Enzyme inhibition, Enzyme activation, Global data fitting, Model comparison, Drug development, Inhibition constant

Funding: The authors received no funding for this work.

==============================
The classical way of defining enzyme inhibition has obscured the distinction between inhibitory effect and the inhibitor binding constant. This article examines the relationship between the simple binding curve used to define biomolecular interactions and the standard inhibitory term (1 + ([I]∕Ki)). By understanding how this term relates to binding curves which are ubiquitously used to describe biological processes, a modifier equation which distinguishes between inhibitor binding and the inhibitory effect, is examined. This modifier equation which can describe both activation and inhibition is compared to standard inhibitory equations with the development of global data fitting templates in Excel and via the global fitting of these equations to simulated and previously published datasets. In both cases, this modifier equation was able to match or outperform the other equations by providing superior fits to the datasets. The ability of this single equation to outperform the other equations suggests an over-complication of the field. This equation and the template developed in this article should prove to be useful tools in the study of enzyme inhibition and activation.

Introduction

The historical development of enzyme-inhibitory theory relied on the generation of rapid equilibrium inhibitory equations akin to the derivation of the Michaelis–Menten equation. These equations developed inhibitory theory around a single constant, termed the inhibition constant (Ki), which when inserted into the Michaelis–Menten equation (Eq. 1; Michaelis & Menten, 1913), in various ways, was used to describe apparent shifts in measured values of the maximum reaction rate (Vmax) and the Michaelis constant (KM) (McElroy, 1947). (1) v=SS+KMVmax.

The Michaelis–Menten equation (Eq. 1) shares the same mathematical structure as the Hill-Langmuir equation (Eq. 2) or ligand–receptor binding relationship (Eq. 3; Gesztelyi et al., 2012). The main difference is that the Michaelis–Menten equation describes the rate of catalytic turnover by an enzyme, where chemical bonds are broken or formed, rather than strictly molecular associations such as the binding between ligand and receptor (Eq. 3) or the binding of molecules to a surface as in the case of the Hill-Langmuir equation (Eq. 2). (2) θ=LnLn+Kd

(3) Receptor binding=LL+Kd.

These equations all take the same form, relating a change in response or signal (v, ϕ, receptor binding), to the concentration of a substance ([S], [L]) based on a constant (KM, Kd) that is itself defined as a concentration of that substance. For example, in the Michaelis–Menten equation, the fraction of the total possible enzymatic conversion of substrate to product (v) is determined by the substrate binding affinity, the Michaelis constant (KM). The substrate binding affinity is the concentration at which the reaction velocity (v) is half that of the theoretical maximum reaction rate (Vmax). This relationship can be easily demonstrated by assuming that an enzyme with a KM value of 1 is exposed to a substrate concentration of 1 ([S] = 1). This produces the situation where the substrate concentration of 1 is divided by itself plus the KM value of 1, yielding the Vmax multiplied by 12. This association produces the hyperbolic relationship between compound concentration and response ubiquitously found in equations used to describe biological interactions (Fig. 1A). The simple relationship is derived from chemical equilibrium mass action relationships and in general, governs most interactions at the molecular level. This relationship has even been used to distill inhibitory theory down to its most basic form, IC50 values (Sebaugh, 2011; Eq. 4), where the inhibitory binding constant is denoted as the concentration of inhibitor needed to reduce the target enzyme’s activity by 50%.

(4) %Inhibition=II+IC50×100

IC50 values are the most common way of characterizing inhibitors, as they provide an easy way of comparing the inhibitory potential of compounds being developed as new drug candidates. IC50 values however only describe changes in the enzyme’s reaction rate (v) and are not an indication of variations in the maximal turnover (Vmax) or substrate affinity (KM).

Figure 1 Enzyme-substrate-modifier interactions.

(A) Enzyme-substrate binding, like any bimolecular system where ligand is in excess, can be expressed using a hyperbolic binding curve. Similarly, hyperbolic binding curves are also useful for describing the binding of modifiers, either inhibitors or activators, with the enzyme. (B) A basic way of conceptualizing the rate at which an enzyme population hydrolyses its substrate and how that rate may be affected by modifiers, is to limit the potential states the enzymes may be found in to free enzyme, enzyme-substrate complex, enzyme-modifier complex and enzyme-substrate-modifier complex. Catalysis is then defined by the portion of the substrate bound population affected by modifier (kcat2) or free of modifier (kcat1). (C) The hyperbolic association of substrate (yellow boxes) and modifier (blue boxes) with the enzyme population is then able to provide a way of determining the rate of substrate catalysis. The depicted table is very similar to a simple multiplication table where the percent of substrate associated enzyme is displayed vertically with yellow bars, while association of modifier is displayed horizontally with blue bars. Overlap of the two populations is depicted as green, and along with the yellow bars represent the portion of the enzyme population which are catalytically relevant. While the hyperbolic curves described by the binding isotherm is a continuum between 0% and 100% association, the table is limited to 0%, 25%, 50%, 75% and 100% for simplicity. Substrate hydrolysis is then defined by the portion o the enzyme population associated with substrate in the presence or absence of modifier. For example, in the absence of modifier (0%), at a substrate concentration equal to the KM, 50% of the enzyme population is bound by substrate and the reaction rate is half that of the VMAX1. However, if a concentration of modifier equal to the modifier binding constant (KX) is added, half of the enzyme population is shifted to the new catalytic rate (kcat2) and substrate affinity (KM2). This results in 25% of the population hydrolysing substrate free of modifier (Yellow box) and 25% shifted to the altered state (green box). The altered state produced by the modifier may result in a very different substrate association than that observed with the unmodified enzyme population, so it must be recognized that the green boxes represent the portion of the population that is altered by the modifier unlike the yellow boxes that represent substrate association and can be directly related to the VMAX1.

Traditionally, changes in reaction velocity produced by changes in substrate affinity and/or maximal velocity, have been defined with equations that were derived from reaction schemes based on enzyme, substrate and inhibitor interactions. This method of describing enzyme inhibition was highly dependent on the use of inhibition constants (Ki) which initially made its appearance in the competitive (Eq. 5), non-competitive (Eq. 6), uncompetitive (Eq. 7) and mixed non-competitive inhibition equations (Eq. 8) (McElroy, 1947; Cleland, 1970).

(5) v=SS+KM1+IKiVmax

(6) v=SS+KM1+IKiVmax

(7) v=SS1+IKi+KMVmax

(8) v=SS1+IαKi+KM1+IKiVmax

While these equations added both inhibition constants and terms for the inhibitor concentration to the Michaelis–Menten equation, absent are terms defining the potential catalytic activity of the enzyme-inhibitor complex. This may be due to the mechanisms used in the derivation of these equations which do not take into account partial inhibition and have resulted in their designation as total inhibitors (Cleland, 1970). To overcome this limitation, other equations have been developed to describe compounds that do not completely stop the catalytic activity of their target (Bisswanger, 2002; Cleland, 1970; Segel, 1975; Yoshino, 1987). However, these equations, known as partial inhibition equations, are rarely utilized in the literature.

So what do the equations for total inhibition describe? An easy way of visualizing how these equations are believed to affect the activity of an enzyme is to plot experimentally determined values of Vmax and KM on a Cartesian coordinate graph with Vmax on the y-axis and KM on the x-axis (Fig. 2A). If the catalytic activity of an enzyme is defined as the coordinates KM and Vmax then inhibtion or activation of the enzyme’s activity can be expressed as a shift to a different position on the graph. For example, the classical competitive inhibition equation (Eq. 5) represents a decrease in substrate binding resulting from the presence of a substrate mimic that blocks the enzymes active site. This is characterized by a decrease in apparent substrate affinity producing an increase in the apparent KM value from its initial value to infinity in a linear fashion (Fig. 2B). While, the non-competitive inhibition equation (Eq. 6), represents a hyperbolic decrease in Vmax from its initial value to zero (Fig. 2C). The uncompetitive equation (Eq. 7) causes an apparent reduction in the KM value implying a higher substrate affinity, while also decreasing the apparent value of the Vmax (Fig. 2D). The mixed non-competitive inhibition equation (Eq. 8) produces a reduction in the Vmax while either increasing or decreasing the KM based on the ratio between Ki and Kiα (Fig. 2E). The changes in enzymatic activity described by these equations leave many other undefined inhibitory and stimulatory possibilities (Fig. 2F). As previously stated, while these equations are the most common forms of inhibition reported in the literature, aside from IC50s, their primary disadvantage is their inability to describe the activity of an enzyme-inhibitor complex. This has been addressed with the derivation of separate sets of equations to cover what is referred to as the partial forms of inhibition associated with each of the classical inhibition types, i.e., partial competitive, partial non-competitive, partial uncompetitive and partial mixed non-competitive (Bisswanger, 2002; Cleland, 1970; Segel, 1975; Yoshino, 1987). To simplify and standardize the field Fontes, Ribeiro & Sillero (2000) and more recently Baici (2015) have attempted to redefine all the possible interactions inhibitors and activators may have with an enzyme. However, as the complexity of the proposed equations has continued to increase, their application has trailed off, with many journals now accepting or having a preference for IC50 values (Brandt, Laux & Yates, 1987; Lazareno & Birdsall, 1993).

Figure 2 Cartesian coordinate plots.

Cartesian coordinate plots of (A) the maximum velocity (Vmax) and substrate affinity constants (KM) used to define the Michaelis–Menten (B) the effect of competitive inhibition (C) non-competitive inhibition and (D) mixed non-competitive inhibition. (E) A representation of the full range of effects which may occur using Eq. (13) and (F) A plot of a theoretical compound which activates the catalytic rate while decreasing substrate affinity emphasizing the hyperbolic relationship that should govern a transition between any of the points on the Cartesian plot.

In my opinion, overcomplication of the enzyme modifier kinetics is contributing to the demise of the field and this overcomplication is related to the treatment of Ki in the total inhibitor equations (Eqs. 5–8). In the total inhibitor equations, the Ki is equated to the effect of the inhibitor on the enzymatic activity rather than an equilibrium binding constant marking the concentration where half the enzyme population is bound by the inhibitor.

The arrangement of the Ki in the total inhibition equations is unusual, in that, while the general term (Eq. 9) appears to be the same in all of the equations (Eqs. 5–8), it functions as a factor of the denominator in the non-competitive equation (Eq. 6) and as a factor of individual terms in the denominator with the other equations (Eqs. 5–(8)). Additionally, this general term (Eq. 9) that is supposed to describe the binding of the inhibitor to the enzyme does not share the same format as other equations used to describe biological interactions (Eqs. 1–4). However, a rearrangement of the non-competitive equation (Eq. 10) demonstrates that this notation is actually the reciprocal form of the hyperbolic equation used to describe biological interactions (Eq. 11; Walsh, 2012). (9) 1+IKi

(10) v=SS+KM1+IKiVmax=SS+KMVmax−VmaxII+Ki

(11) 11+IKi=1−II+Ki.

This rearrangement (Eq. 10), directly relates the non-competitive equation’s hyperbolic decrease in Vmax, to the binding of the inhibitor with the enzyme population. This rearrangement also explains why the non-competitive inhibition equation is limited to situations where the inhibitor completely stops the catalytic activity of the enzyme, as the Vmax is reduced by itself, as the inhibitor binds the enzyme population (Eq. 10). This alternate form of the inhibitory term also suggests a rationale for the odd pattern of the classic competitive inhibition equation. In the competitive equation (Eq. 5), the KM is multiplied by the inhibitory term (Eq. 9) resulting in the KM getting divided by the fraction of the enzyme population not bound by the inhibitor (Eq. 12). This produces the linear trend of increasing KM driving its value to infinity rather than generating a hyperbolic shift from one substrate affinity to another. A one to one association of inhibitor with enzyme would mean that each enzyme bound by inhibitor expresses the new apparent KM value induced by the inhibitor. As the enzyme population is converted from an inhibitor-free group to an inhibitor bound group, the observed KM would shift from the initial KM to the inhibitor-induced apparent KM in a hyperbolic manner. Therefore, the competitive model cannot describe changes in KM resulting from a one to one association of the inhibitor with the enzyme. (12) v=SS+KM1+IKiVmax=SS+KM1−II+KiVmax.

While many inhibitors that only change substrate affinity are classified as competitive, it is not hard to envision situations where changes in enzyme-substrate binding could be caused by interactions not related to blockage of the enzyme’s active site by an inhibitor which mimics the substrate. For example changes in the conformation of the active site could reduce the ability of the substrate to bind without reducing the catalytic rate of the enzyme. This could occur through alosteric interactions or even through partial blockade of the active site when the enzyme is associated with the inhibitor. For example, the peptidase kallikrein was believed to be competitively inhibited by benzamidine (Sousa et al., 2001). However, the crystal structure of benzamidine binding to kallikrein demonstrated that it does not block the catalytic site of the enzyme but instead binds to a portion of the protease that deals with substrate specificity. Known as the side chain binding pocket, benzamidine binds to a portion of the enzyme which recognizes the side chain of phenylalanine (Bernett et al., 2002). This results in a hyperbolic decrease in substrate affinity based on the portion of the kallikrein population bound to benzamidine. While each kallikrein enzyme bound by the benzamidine has less affinity for its substrate it still hydrolyses the substrate at the same rate. This is supported by a better fit of the experimental data to a hyperbolic rather than linear change in KM (Walsh, Martin & Darvesh, 2011a).

While inhibitor interactions that conform to the traditional competitive equation cannot be ruled out, the evidence for classifing an inhibitor as competitive must be closely scrutinized before the inhibition can be attributed to the standard competitive equation (Eq. 12).

Assuming that enzyme-inhibitor interactions are dependent on the same relationship which defines other molecular systems (Eqs. 1–4), the Michaelis–Menten equation can be modified to accommodate both positive and negative changes in KM and Vmax by adding terms which relate binding of the inhibitor with the enzyme population directly to change in enzymatic activity (Walsh, Martin & Darvesh, 2007; Eq. 13). (13) v=SS+KM1−KM1−KM2XX+kxVmax−Vmax−VmaxXX+kx

In this equation, the changes from the initial KM and Vmax values are directly related to the binding of modifier (X) with the enzyme (Figs. 1B, 1C). The change from inhibitor to modifier notation refers to the ability of this equation to describe activators of enzymatic activity as well as inhibitors. The numrical subscripts associated with the Vmax and KM are used to represent the distinct states of the enzyme. For example in the absence of modifier the Vmax and KM are denoted as V12ptmax1 and KM1 while V12ptmax2 and KM2 represent Vmax and KM values produced by the modifier. By clearly defining V12ptmax2 and KM2, this equation can be used to model either negative or positive changes in the Vmax and KM (Fig. 2F) provided the shifts are hyperbolic. As previously stated the designation of a V12ptmax2 stems from a simple rearrangement of the non-competitive inhibition equation (Eq. 10), while the term describing changes to the KM can be derived the same way the other classical equations have been derived, using the rate equation, conservation of mass and equilibrium relationships (Supplemental Information 1). Indeed, the main failing of this equation may be that it is unable to produce the linear increase in KM which characterizes the standard competitive inhibition equation (Fig. 2B). However, whether previously observed linear changes in KM are in fact linear or just represent the linear portion of a hyperbolic curve, (as it could be argued was the case with benzamidines’ inhibition of kallikrein) deserves more attention (Walsh, Martin & Darvesh, 2011a).

Materials & Methods

Templates for comparing inhibitor and activator equations were developed using Excel. All enzyme kinetic data analyzed in this study was collected from previously published results or simulated using the equations described. The ability of the equations to model the data was evaluated using non-linear regression with the solver add-in of Excel to globally fit the data (Kemmer & Keller, 2010).

Results & Discussion

To truly assess the fitting of an equation to experimental data the equation should be globally fit to the data. To this end, a template which can compare the capacity of the classical inhibition equations (Eqs. 5–8) and the modifier equation (Eq. 13), to globally fit experimental data was developed (Supplemental Information 2). To illustrate the functionality of the template, data was acquired from Biotek’s application note on basic enzyme kinetic determinations (Held, 2007), where the inhibition of β-galactosidase by β-D-thiogalactopyranoside was examined. The structural similarity between the inhibitor and the substrate, combined with the pattern observed using a double reciprocal plot lead to the conclusion that β-galactosidase was competitively inhibited by β-D-thiogalactopyranoside (Held, 2007). However, this analysis was based on standard pattern recognition where regression lines for each inhibitor concentration were overlaid and convergence of the lines close to the y-axis was interpreted as competitive inhibition. This sort of analysis does not determine whether the pattern produced by the regression lines conforms to a global fitting of the competitive inhibition equation (Eq. 12) to the experimental data. Indeed, this reliance on pattern recognition is a major hindrance for proper identification of inhibition mode. To address this issue, the template has been designed to facilitate the quick comparison of the non-competitive, competitive, uncompetitive, mixed non-competitive and modifier equation (Eqs 5–8 & 13, Fig. 3, Supplemental Information 2, Please refer to Supplemental Information 3 for step by step pictorial instructions on the use of the fitting template). To determine if the data from Biotek’s application note truly does conform to the classical competitive inhibition model the data was analyzed using the modifier template (Fig. 3A). Inserting data into the template generates KM and Vmax values (Fig. 3B) using a modified direct linear plot. The modified direct linear plot provides a statistically robust way of determining apparent KM and Vmax values by providing N(N − 1)∕2 intercept values from which the median can be determined (Cornish-Bowden, 1995). These median values are used as initial parameters in the fitting of the various inhibition equations. The KM and Vmax generated by the modified direct linear plot are in close agreement with the values reported by Held (2007), calculated KM 0.15 mM Vmax 28.2 mOD/min versus reported KM 0.24 mM Vmax 33.4 mOD/min. Additionally, the template provides a Ki estimate based on the decrease in observable rate associated with the top substrate concentration ([S]1) and the assayed inhibitor concentrations ([I]1 to [I]7, Fig. 3A). The fit of the inhibition equations using the initial kinetic parameter is displayed both tabularly and graphically (Fig. 3C). The primary table contains the parameters employed in the fitting of each equation and values used to assess the ability of each equation to model the data. The columns containing values to evaluate the fit, namely the sum of squared residuals (RSS), relative standard error (RSE) for the regression and the Bayesian information criterion (BIC), which are color-coded such that the smallest values appear green representing the best fit and red the worst. These parameters allow evaluation of the ability of each equation to fit the observed data set with the Bayesian information criterion being included for evaluation of potential overfitting as it negatively scores fittings based on the complexity (number of parameters) of the model being used (Burnham & Anderson, 2002). In this case, the number of parameter for each model is listed in the table as k. Representation of the fit of each equation is also visualized with a boxplot of the residuals, with the residuals used to generate the boxplot appear to the right of the corresponding boxplot. Ideally, a good fit would consist of an even distribution of the residual values around zero so for evaluation purposes a secondary table is presented which contain values used in the generation of the boxplot. The initial parameters produced by the template may result in fairly good fits or extremely poor fits as is apparent in the poor distribution of the residuals with the modifier equation (Fig. 3C).

Figure 3 Enzyme modifier template.

The enzyme modifier kinetic template (A) provides fifteen rows for substrate concentrations as well as sixteen columns for varying concentrations of enzyme modifiers, either activators or inhibitors. (B) Below the raw data, a modified direct linear plot uses the data in the no inhibitor column to generate estimates of the KM and Vmax while the first row of data is used to produce a linear estimate of the initial Ki value. (C) The initial kinetic values are inserted into a table which contains the parameters utilized in the fitting of each equation covered by the template. The table also contains the Sum of Squared Residuals (RSS) and the Bayesian information criterion (BIC) for assessing the fit of the model based on the provided parameters. Additionally, a box plot of the residuals is provided to offer a visual representation of the error associated with the fitting of each equation to the data.

To apply a global fit to the data the solver add-in for Excel is utilized (Kemmer & Keller, 2010), Please refer to Supplemental Information 3 for step by step instructions on using the solver feature with the template). In fitting to the Biotek data, the solver feature was used to minimize the RSS of the fits, initially by varying parameters for the inhibition followed by all the parameters associated with the equation. For example, the fitting of the non-competitive inhibition equation was performed by minimizing the RSS through varying the Ki value, this was followed by a second minimization of the non-competitive RSS value by varying the Vmax, KM and Ki simultaneously.

The improvement in fit between the initial parameters generated by the template (Fig. 3C) and those present after minimizing the residuals is clear (Fig. 4, Supplemental Information 4). Both RSS and BIC values are noticeably reduced and the boxplot demonstrates a much evener distribution of the residuals around zero (Fig. 4A). The presented values suggest that rather than β-D-thiogalactopyranoside conforming to the classical competitive inhibition model a better fit can be produced using the modifier equation which assumes a hyperbolic change in KM and Vmax. Global fitting of the data with each equation is plotted below the boxplot (Figs. 4B–4F) For each equation, the data is presented as a correlation plot of the calculated versus the experimental data, an overlay of the model with the experimentally observed rates (v vs [S]), a double reciprocal Lineweaver-Burk plot (1/v vs 1/[S]) (Lineweaver & Burk, 1934) and a Dixon plot (1/v vs [I]) (Dixon, 1953; Butterworth, 1972). The correlation plot provides another way of visualizing the ability of each equation to fit the data as a linear regression of the observed versus the calculated values should produce a slope of 1 and a high R2 value if the calculated values equal the observed values. For the Lineweaver-Burk plots, the lines represent the overlay of the globally fit equations rather than best fit linear regressions of the individual data sets.

Figure 4 Inhibition of β-galactosidase by β-D-thiogalactopyranoside.

Global fitting of the Biotek’s application note data (Held, 2007) to multiple inhibitory equations. (A) In addition to producing global minimal fitting values based on the RSS, the modifier template also produces a visualization of the fitting of each inhibitory model with correlation plots of the experimental and calculated values, double reciprocal Lineweaver-Burk plots, direct plots of the reaction rate versus the substrate and Dixon plots. Shown are the global fits of the (B) Non-competitive (C) Competitive (D) Uncompetitive (E) Mixed Non-competitive and (F) The modifier equation to the data.

An examination of the competitive plots (Fig. 4C) demonstrates the deviation of the observed data from the competitive model, where the model at higher inhibitor concentration and lower substrate concentration suggests lower rates than those observed. This problem is mirrored by the mixed non-competitive equation (Fig. 4E) which approximates the linear increase in KM produced by the competitive equation as long as the predicted αKi is significantly removed from the range of the Ki value, as is observable in the fitting (Ki = 4.2 ×10−4 and αKi = 1.4 ×10−1, Fig. 4A). As previously stated the modifier equation (Eq. 13) provides a better fit to the data which is apparent specifically in the low substrate, high inhibitor region of the Lineweaver-Burk plot (LWB plot Top Line Fig. 4F) and the high inhibitor region of the Dixon plot. Unfortunately, the Ki for the fit produced in Biotek’s application note was not provided so a more in-depth comparison of the templates ability to fit the data cannot be undertaken.

A more thorough evaluation of the present method can be realized by studying a recent publication by Pintus et al. (2015) which describes the discovery of E. characias leaf extracts with tyrosinase inhibitory activity. The inhibitory properties of these extracts were characterized using Lineweaver-Burk plots and the data used in their analysis was made available online. To determine if the data conformed to their reported modes of inhibition, the data provided in their supplementary information was analyzed using the template. The Lineweaver-Burk plot of their aqueous extract suggested that it acted as a mixed non-competitive inhibitor. This analysis was not based on global fitting of the model to the data but rather the accepted pattern recognition associated with the position of the intercept produced by the individual best fit linear regression lines for the data produced with varying inhibitor concentrations. From the best fit linear regression lines, the Ki and αKi were reported as 0.097 and 0.33 mg/mL. Using a global fitting approach produced slightly different values (0.099 and 0.37 mg/mL) and almost halved the associated RSS value (RSS 7 ×10−4 to 4 ×10−4 Fig. 5A, Supplemental Information 5). Global fitting agreed with the reported inhibition model suggesting that only the mixed non-competitive (Fig. 5B) or modifier (Fig. 5C) equations were able to adequately model the data.

The Lineweaver-Burk plot of their ethanolic extract was reported to produce the recognizable competitive inhibition pattern where the linearly regressed best-fit lines intercepted on the Y-axis (Pintus et al., 2015). However, when the data was examined using global fitting, the competitive model did not demonstrate a significantly better fit to the data when compared to the other models. When the reported Ki (23.7 µg/mL) was fixed during the global fitting process the sum of squared residuals was further worsened (RSS 0.0183 vs. 0.0143, Fig. 6A & Supplemental Information 6). Compared to the other models, the only fit which was worse than the competitive model was the uncompetitive form of inhibition. Even the non-competitive model which was completely unable to model the results of the higher inhibitor concentrations was able to produce a slightly better fit according to the sum of squared residuals (Figs. 6A–6C). This is a good example of the limitations associated with the competitive model, as the mandatory linear increase in KM described by the model, requires a pattern with a strict distribution of the lines in a double reciprocal Lineweaver-Burk plot rather than simply an intercept on the Y-axis. As is apparent, in the Lineweaver-Burk plot, global fitting of the competitive equation produced a relatively good fit to the data in the absence of inhibitor (lowest line in LWB plot Fig. 6C) and to the data for the enzyme in the presence of the highest concentration of inhibitor (highest line in LWB plot Fig. 6C). However, the other lines of the plot are clearly above the data points that they should be bisecting for a proper fit. For this situation, global fitting suggests that the mixed non-competitive and modifier models both provide better fits than the competitive equation (Figs. 6A, 6D, 6E).

Figure 5 Tyrosinase inhibition by E. characias aqueous extract.

Global fitting of the E. characias aqueous leave extract reported as a mixed-non-competitive inhibitor of tyrosinase (Pintus et al., 2015). (A) Fitting suggests the modifier and mixed non-competitive equations model the data significantly better than the other equations. Shown are the global fits of the (B) Mixed Non-competitive and (C) the modifier equation to the data.

Figure 6 Tyrosinase inhibition by E. characias ethanolic extract.

Global fitting of the E. characias ethanolic leave extract reported as a competitive inhibitor of tyrosinase (Pintus et al., 2015). (A) Fitting suggests the modifier and mixed non-competitive equations model the data better than the other equations. Shown are the global fits of the (B) Non-competitive (C) Competitive (D) Mixed Non-competitive and (E) the modifier equation to the data.

Partial Inhibition Equations

The limitations of the total inhibition equations have been acknowledged through the development of partial inhibition forms for each of these equations, ie., the partial non-competitive (Eq. 14; Segel, 1975), partial competitive (Eq. 15; Segel, 1975), partial uncompetitive (Eq. 16; Bisswanger, 2002) and partial mixed non-competitive (Eq. 17; Yoshino, 1987).

(14) v=VmaxSKs+βVmaxSIKsKi1+SKs+IKi+SIKsKi

(15) v=VmaxSKs+SαKsKi1+SKs+IKi+SIαKsKi

(16) v=Vmax+V2IKiSKs+1+IKiS

(17) v=Vmax1+IKi′βSKs1+IKi′SKs+1+IKi′Ks′Ks1+IKi′.

While there has been limited use of these equations where the raw data is accessible, Whiteley (1997), Whiteley (1999), Whiteley (2000) expanding on Yoshino’ (1987) work identifying forms of partial inhibition through the examination of fractional velocity plots, made the data in his papers available. The modifier template developed in the previous section also has the advantage that almost any equation can be easily inserted into the spreadsheets for global fitting analysis. This allowed the global fitting of the data presented by Whiteley (1997), Whiteley (1999), Whiteley (2000) to be analyzed with the total inhibition (Supplemental Information 2) and the partial inhibition equations using a version of the template modified to model the partial inhibition (Supplemental Information 7).

In Whiteleys’ article examining partial competitive inhibition, data for the inhibition of glutamine synthase by alanine was presented as an example of this form of inhibition (Whiteley, 1997). Inserting the data into the modifier template suggests that the data did not conform to the traditional inhibitory equations, but was modeled by the modifier equation very well (Fig. 7A, Supplemental Information 8). Fitting the data to the partial inhibition equations did indicate that partial competitive inhibition provided an even distribution of the residuals and a slightly better fit than the competitive inhibition model. However, of the partial inhibition models, the partial mixed non-competitive inhibition equation (Eq. 17) was the only model able to fit the data as well as the modifier equation (Fig. 7B, Supplemental Information 9).

Figure 7 Putative partial competitive inhibition.

Global fitting of the data presented in Whiteleys’ article on partial competitive inhibition (Whiteley, 1997) to (A) the modifier equation and the classical inhibitory equations, and (B) the modifier equation and the partial inhibitory equations.

In a subsequent publication on partial and complete non-competitive inhibition, Whiteley provides two examples of inhibition. The first example of inosine nucleosidase inhibition by adenine is presented as a partial non-competitive form of inhibition and the second example in which adenosine monophosphate is used to inhibit alcohol dehydrogenase is classified as non-competitive (Whiteley, 1999).

Examining the first example suggests that none of the basic models fit the data as well as the modifier equation (Eq. 13; Fig. 8A; Supplemental Information 10). When examined with the partial inhibition template the partial non-competitive (Eq. 14) and partial mixed non-competitive (Eq. 17) equations provided slightly better fits than the total inhibition models but were unable to improve on the fit provided by the modifier equation (Fig. 8B, Supplemental Information 11).

Figure 8 Putative partial and complete non-competitive inhibition.

Global fitting of the partial non-competitive data presented in Whiteleys’ article on partial and complete non-competitive inhibition (Whiteley, 1999) to (A) the modifier equation and the classical inhibitory equations, and (B) the modifier equation and the partial inhibitory equations. Global fitting of the non-competitive data presented in Whiteleys’ article to (C) the modifier equation and the classical inhibitory equations, and (D) the modifier equation and the partial inhibitory equations.

In the second example, rather than presenting as non-competitive the fitting suggested that the modifier, mixed non-competitive and partial mixed non-competitive equations all provided improved and roughly equivalent fits to the data (Figs. 8C, 8D; Supplemental Information 12, 13).

Whiteleys’ most recent publication on identifying partial forms of inhibition, identifies adenosine triphosphate as a partial uncompetitive inhibitor of mevalonate diphosphate decarboxylase (Whiteley, 2000). However, when globally fit to the total and partial inhibition equations, even the uncompetitive inhibition equation outperforms the partial uncompetitive equation (RSS:1.69 ×10−5 vs. 1.92 ×10−2, Figs. 9A, 9B; Supplemental Information 14, 15). Out of all the models, the partial uncompetitive fared the worst while the modifier equation and the partial mixed non-competitive equation modeled the data the best (Fig. 9B).

Figure 9 Putative partial uncompetitive inhibition.

Global fitting of the data presented in Whiteleys’ article on partial uncompetitive inhibition (Whiteley, 2000) to (A) the modifier equation and the classical inhibitory equations, and (B) the modifier equation and the partial inhibitory equations.

Overall equation fitness

A comparison of the ability of the equations to fit the examined experimental datasets suggests that the modifier equation (Eq. 13) can fit each example just as well if not better than all the other equations (Table 1, Supplemental Information 16–18). Indeed, only the partial mixed non-competitive equation (Eq. 17) was comparable to the modifier equation in its ability to fit the experimental datasets. The ability of the modifier equation to outperform the other equations was further supported with simulated data (Table 2). Using simulated data for the non-competitive (Eq. 6, Supplemental Information 19, Supplemental Information 20), competitive (Eq. 5, Supplemental Information 21, 22), uncompetitive (Eq. 7, Supplemental Information 23, 24), mixed non-competitive (Eq. 8, Supplemental Information 25, 26) equations and an example of activation generated with the modifier equation (Eq. 13, Supplemental Information 27, 28), the ability of each of the models to fit the simulated data was also examined. The simulated data contained many more data points than the experimental data used in the fittings found in Table 1. This highlighted the inability of the total inhibitor equations aside from the mixed non-competitive inhibition equation to model the data generated with the other total inhibitor models. For example, the competitive equation was unable to fit the data produced with the non-competitive equation (Table 2, RSS 3100). The modifier equation, apart from the competitive inhibition simulated data, was able to fit the other simulated data sets as well as or better than the other equations. Similarly, the partial mixed non-competitive equation also produced a good fit for the datasets and was able to fit the example of activation generated with the modifier equation (Table 2, Supplemental Information 28). This suggests the partial mixed non-competitive equation may be almost as adaptable as the modifier equation for describing a wide variety of modifier interactions. However, the modifier equation outperformed the partial mixed non-competitive equation in all the simulated datasets.

Table 1 Comparison of experimental data fitting between equations.

RSS values related to the global fitting of the literature datasets (Supplemental Information 4–6, 8–18) with the equations in the templates (Eqs. (5)–(8), (13)–(17)). For each literature dataset, the reported mode of inhibition is listed in the left-hand column and is circled in the table. The ability of each model to fit the datasets have been color-coded such that superior fits appear in green with the text of minimum RSS values appearing in red.

	Non-competitive	Competitive	Uncompetitive	Mixed non-competitive	Modifier equation	Partial non-competitive	Partial competitive	Partial uncompetitive	Partial mixed non- competitive	
Competitive (Held, 2007)	3.10E+02	3.15E+01	4.30E+02	3.07E+01	2.28E+01	2.65E+02	5.85E+01	2.42E+03	2.28E+01	
Mixed non-competitive (Pintus et al., 2015)	3.90E−03	6.90E−03	2.02E−02	4.36E−04	3.62E−04	3.76E−03	5.00E−04	9.85E−02	3.62E−04	
Competitve (Pintus et al., 2015)	1.39E−02	7.63E−03	1.60E−01	7.57E−03	7.57E−03	1.39E−02	7.63E−03	1.60E−01	7.57E−03	
Partial competitive (Whiteley, 1997)	8.63E−04	3.22E−04	5.33E−03	3.22E−04	1.09E−06	5.41E−04	3.22E−04	7.46E−03	1.09E−06	
Partial non-competitive (Whiteley, 1999)	5.54E−04	1.49E−03	1.64E−03	5.49E−04	1.55E−04	1.62E−04	5.49E−04	2.07E−02	1.62E−04	
Non-competitive (Whiteley, 1999)	3.74E−06	1.33E−03	1.92E−03	3.13E−06	3.13E−06	3.74E−06	3.16E−06	6.59E−03	3.13E−06	
Partial uncompetitive (Whiteley, 2000)	1.16E−04	3.18E−04	1.69E−05	1.69E−05	1.97E−06	1.01E−04	1.73E−05	1.98E−02	1.98E−06	

Table 2 Comparison of simulated data fitting between equations.

RSS values related to the global fitting of the simulated datasets (Supplemental Information 19– 28) with the equations in the templates ((5)–(8), (13)–(17)). The RSS value of the equations used to generate the dataset has been omitted. The ability of each model to fit the datasets have been color-coded such that superior fits appear in green with the text of minimum RSS values appearing in red.

	Non-competitive	Competitive	Uncompetitive	Mixed non-competitive	Modifier equation	Partial non-competitive	Partial competitive	Partial uncompetitive	Partial mixed non- competitive	
Non-competitive		3.10E+02	5.84E+01	8.55E−10	2.08E−28	1.19E−28	1.54E−05	8.60E+02	4.44E−08	
Competitive	9.59E+01		2.25E+02	1.91E−07	1.33E−03	9.52E+01	2.26E−06	3.27E+02	1.57E−01	
Uncompetitive	2.17E+01	4.16E+02		6.14E−07	6.13E−08	2.17E+01	4.16E+02	6.82E+02	1.46E−07	
Mixed non-competitive	8.21E+01	1.24E+00	2.08E+02		7.23E−08	8.15E+01	9.78E−04	3.40E+02	2.12E−04	
Modifier equation (activation)	6.29E+02	6.29E+02	6.29E+02	6.29E+02		6.29E+02	4.72E+04	2.88E+02	1.81E−06	

Conclusions

Based on these examples, the modifier equation (Eq. 13) has been able to model each dataset just as well if not better than the other equations based on the sum of squared residual values. While both the inhibition of β-galactosidase by β-D-thiogalactopyranoside (Held, 2007) and inhibition of tyrosinase with an ethanolic extract of E. characias leaves (Pintus et al., 2015) were reported as examples of competitive inhibition, global fitting of their data suggested they do not conform to the classical competitive inhibition equation (Figs. 4 & 6). As none of the datasets conform to a linear change in KM, it is not surprising that the modifier equation which directly relates fractional association of modifiers with the enzyme population to change in activity fits all the examples very well.

The modifier equation defined here unifies inhibition and activation in a single equation by describing changes in Vmax and KM using a single binding constant (Eq. 13), something which was not described with the traditional equations such as the mixed non-competitive equation (Eq. 8). The clear distinction between inhibitor binding constants and effect on KM and Vmax also permits the modular expansion of the Michaelis–Menten equation to accommodate multiple substrate and modifier binding interactions (Walsh, 2012). This approach has already proven its value, providing valuable new insight into how the compound DAPT interacts with the multiple-substrate regulated forms of γ-secretase and the implications this has for amyloid precursor protein processing in Alzheimer’s disease (Walsh, 2014). Additionally, it has been used to provide more information on the effect drugs for Alzheimer’s disease have on the multiple-substrate regulated forms of cholinesterases (Walsh, Martin & Darvesh, 2007; Walsh et al., 2011b).

New initiatives for reproducibility and openness such as the database proposed by the Standards for Reporting Enzyme Data (STRENDA) commission which will include raw data (Tipton et al., 2014) suggests enzyme kinetic data will become much more transparent. This transparency will allow easier sharing and evaluation of raw data sets, which will in turn lead to the refitting of raw data with alternative models such as the modifier equation. The global fitting templates presented here should be useful for both evaluating model suitability and in assessing whether the modifier equation described here can replace traditional approaches to inhibition and activation modeling.

Supplemental Information

Supplemental Information 1 Derivation

Click here for additional data file.

Supplemental Information 2 Modifier template

Click here for additional data file.

Supplemental Information 3 Template data fitting

Click here for additional data file.

Supplemental Information 4 Gen5 Biotek inhibition of β-galactosidase by β-D-thiogalactopyranoside

Click here for additional data file.

Supplemental Information 5 E. characias leave aq extract tyrosinase inhibitory activity

Click here for additional data file.

Supplemental Information 6 E. characias leave EtOH extract tyrosinase inhibitory activity

Click here for additional data file.

Supplemental Information 7 Partial inhibition template

Click here for additional data file.

Supplemental Information 8 Modifier template alanine inhibition of glutamine synthase

Click here for additional data file.

Supplemental Information 9 Partial inhibition template alanine inhibition of glutamine synthase

Click here for additional data file.

Supplemental Information 10 Modifier template inosine nucleosidase inhibition by adenine

Click here for additional data file.

Supplemental Information 11 Partial inhibition template inosine nucleosidase inhibition by adenine

Click here for additional data file.

Supplemental Information 12 Modifier template adenosine monophosphate inhibition of alcohol dehydrogenase

Click here for additional data file.

Supplemental Information 13 Partial inhibition template adenosine monophosphate inhibition of alcohol dehydrogenase

Click here for additional data file.

Supplemental Information 14 Modifier template adenosine triphosphate inhibition of mevalonate diphosphate decarboxylase

Click here for additional data file.

Supplemental Information 15 Partial inhibition template adenosine triphosphate inhibition of mevalonate diphosphate decarboxylase

Click here for additional data file.

Supplemental Information 16 Partial inhibition template Gen5 Biotek inhibition of β-galactosidase by β-D-thiogalactopyranoside

Click here for additional data file.

Supplemental Information 17 Partial inhibition E. characiasacias leave aq extract tyrosinase inhibitory activity

Click here for additional data file.

Supplemental Information 18 Partial inhibition template E. characias leave EtOH extract tyrosinase

Click here for additional data file.

Supplemental Information 19 Noncompetitive inhibition

Click here for additional data file.

Supplemental Information 20 Partial inhibition template noncompetitive

Click here for additional data file.

Supplemental Information 21 Competitive inhibition

Click here for additional data file.

Supplemental Information 22 Partial inhibition template competitive inhibition

Click here for additional data file.

Supplemental Information 23 Uncompetitive inhibition

Click here for additional data file.

Supplemental Information 24 Partial inhibition template uncompetitive

Click here for additional data file.

Supplemental Information 25 Mixed noncompetitive inhibition

Click here for additional data file.

Supplemental Information 26 Partial inhibition template mixed noncomp

Click here for additional data file.

Supplemental Information 27 Activation

Click here for additional data file.

Supplemental Information 28 Partial inhibition template activation

Click here for additional data file.

Additional Information and Declarations

Competing Interests

Author Contributions

Data Availability

The authors declare there are no competing interests.

Ryan Walsh analyzed the data, contributed analysis tools, prepared figures and/or tables, authored or reviewed drafts of the paper, approved the final draft.

The following information was supplied regarding data availability:

The Supplemental Files consist of templates for analyzing enzyme kinetic data and the application of these templates to previously published data and simulated datasets.

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
