# Peer review of "Comparing enzyme activity modifier equations through the development of global data fitting templates in Excel"

_PeerJ, doi:10.7717/peerj.6082_

## Round 0.1 · original submission · Major Revisions

All reviewers considered that your discussion of enzyme inhibition suffered from either lack of clarity (e.g. on whether you mean real Km,Ki,Vmax or apparent Km,Ki,Vmax ) or engagement with the literature. As a result,our manuscript seems to be made of two different parts: a somewhat disjointed introduction, which proposes a rearrangement of enzyme equations to (hopefully) make data interpretation more intuitive, and a second section describing a software tool which does not seem to depend crucially on the rearrangement developed earlier, but rather on comparison of classical inhibition equations to an equation developed by the author and co-workers in 10.1016/j.bbagen.2007.01.001 . (As an aside: That equation seems to be justified in that publication by empirical data fitting, rather than explicitly derived from first-principles (since no overall reaction schemes supporting its derivation are present there) and it is my personal opinion that such a scheme (and ensuing derivation) is absolutely required to ensure that the equation does describes a plausible physical model .)

Among other important points, our reviewers specifically question whether that algebraic rearrangement has scientific merit, and I agree that it does not per se warrant a publication in PeerJ, whose acceptance criteria state that "The submission should be ‘self-contained,’ should represent an appropriate ‘unit of publication’, and should include all results relevant to the hypothesis. Coherent bodies of work should not be inappropriately subdivided merely to increase publication count. " The template spreadsheets themselves, however, were rated very positively by Reviewer #3 , who offers some very specific suggestions to increase their usefulness and applicability. Since bioinformatic software tools may be published in PeerJ (https://peerj.com/about/policies-and-procedures/#discipline-standards) I suggest that you rewrite the manuscript to more clearly focus on the templates, and to include enough data regarding the performance of the templates on the identification of inhibition modes in both noisy and pristine data.

Reviewer 1 ·

Basic reporting

The article compares arranged equations with classical equations of enzyme kinetics. It is not clear what is compared: sometimes the author mentions an arrangement of equations and in other cases he mentions a partial inhibition mechanism. If the problem is the mechanism, he should mention that a equation deduced from a partial inhibition mechanism will sure be better describing the partial inhibition and better when fitting to experimental data where the system is affected by partial inhibition. If this is not the case, it is hard to understand how a arranged equation will fit better than the original. Maybe the author is proposing a new fitting method, however, it was not explicitly mentioned.

Experimental design

There is no description of the experimental design, methods and analysis of the data.

Validity of the findings

The main conclusion was that the modifier equation was able to model in good agreement the experimental data. How obvious is that an arranged equation will fit as well as the original equation? I think that we have two options:
i) The arranged equations are as good as the original equations when used as mathematical models. Unless the author is proposing a new fitting method (which is not explicitly mentioned), the article lacks scientific merit.
ii) The author did not properly nor clearly described the objective of this work to ensure the correct understanding when reading the manuscript.
In addition, as the methods were not described it is difficult to evaluate the quality of the results.

Additional comments

The main objective of the article was not clearly exposed. What the author intend to do? What intend to demonstrate? What is the improvement when using the arranged equations?
The methodology was not described. The methods used to test the ability of the arranged equations to fit experimental data were not described. In the same way, the methods to compare the fitting quality were not described. Indeed, there are a lot of information, calculations and calculation methods implicit in the results that were not described. Surely, a significant improvement of the manuscript is possible adding this information. As an example, the Bayesian information criterion (BIC) was calculated and used to evaluate the data. It is necessary to describe the calculation and usefulness of the BIC in the data evaluation process.
It is necessary to add all this vital information to improve tha manuscript.

Some comments:
1. The author continuously define enzymatic activity as Vmax. This definition is wrong because the enzymatic activity is obtained from an experimental assay in standard conditions which result is not necessarily the Vmax. Indeed, the Vmax can be related to the enzymatic activity, however, the enzymatic activity cannot be related to Vmax. Just in the case that the substrate concentration is saturated (S0 >> Km), the obtained reaction rate will be approximately the Vmax.

2. It is usually found in books and articles the classification of enzymatic inhibition as competitive, non-competitive, uncompetitive and mixed. Exposing the types of inhibition in this way can drive to a misunderstanding because the non-competitive inhibition is a subclass of mixed inhibition. Considering this the types of inhibition has to be exposed as competitive, uncompetitive and mixed inhibition. When the mixed inhibition presents the same value for both inhibition constants, the competitive (Ki) and the uncompetitive (Ki’) components, it means Ki = Ki’, the inhibition is denominated non-competitive. It is important to expose a correct classification of the types of inhibition to finish with the eternal misunderstanding.

3. In line 76 the author indicated that the equations presented are unable to describe partial inhibition. This is correct. However, the inability to describe the partial inhibition is due to the origin of those equations which were obtained from reaction mechanisms describing total inhibition and not partial inhibition. Thereby, the inability in question was wrongly related to those equations when it should be related to the mechanisms proposed to derive those equations.

4. In line 86 the author exposed that the catalytic activity of an enzyme is defined as the coordinates Km and Vmax. As I commented above, the enzymatic activity is obtained from an experimental assay in standard conditions. In this case the author has to talk about “reaction rate”, which is the dependent variable in all the exposed rate equations (v). It is true that the activity is obtained through empirical determination of a reaction rate at certain operating conditions, however, a reaction rate is not necessarily the enzymatic activity.

Reviewer 2 ·

Basic reporting

.

Experimental design

.

Validity of the findings

.

Additional comments

Although the main points made in this paper are correct, the knowledge the author displays of the literature is completely inadequate, as one might guess from the fact that fewer than half of the references are to work done in the 21st century (not counting the author's own publications), and most of those that ARE from the 21st century address points of detail rather than the main thesis. Córtes et al. (2001, Biochem. J. 357, 263-268) provided a more detailed account of the relationship between inhibition constants and IC50, and referred to an earlier study by Cheng and Prusoff (1973), that considered the same question. However, even 2001 is only just in the 21st century, so the author should read Baici's recent book (2015, Kinetics of Enzyme-Modifier Interactions, Springer) as well as reading the relevant papers that he cites.

Sebaugh's paper (2011) is addressed to pharmaceutical scientists, and it is absurd to say that it "distill[s] inhibitory theory down to its most basic form". It contains nothing that a biochemist would call inhibitory theory.

The paper by Tipton et al. (2014) is OK as far as it goes, but it says almost nothing relevant to the present paper. It does, however, include references to work the author should read.

Reviewer 3 ·

Basic reporting

This manuscript presents a package for analyzing enzyme kinetic data by means of global fitting within an Excel template. The author argues that the common practice of characterizing kinetic data by visual examination of graphical representations often leads to oversimplified interpretations. The routines presented in the current work enable the simultaneous analysis of raw data according to standard models of competitive, non-competitive, uncompetitive and mixed inhibition. A second template permits the analysis of the partial versions of these mechanisms. Both templates include an equation (the “Modifier” equation ) incorporating all of the other mechanisms. Vm, Km and Ki values are derived for each mechanism as well as the RSS measure of goodness of fit. The template thus offers a rapid and objective evaluation of different kinetic models and is potentially of considerable value to the field. The Figures in the manuscript mostly consist of clips from the Excel files included in the Supplementary material and are generally helpful summaries of the voluminous information contained in the spreadsheets.
The author argues that the conventional forms of enzyme kinetic equation are too complex and obscure the mode of action of inhibitors. The equation employed for the fitting involves a rearrangement of the more familiar format to demonstrate separate effects of inhibitor or activator on apparent Km or Vm values. To this reviewer, the rearrangement is no simpler than the more traditional forms of the equation. This does not, however, invalidate the result and an alternative view may well be helpful to some.
The design of the work is acceptable. It employs data sets from other workers and subjects these data to the global fitting procedure to compare their fit to the various models employed. At one point in the manuscript, the utility of the process for determining competitive inhibition models is queried. The author might consider testing the validity of the process by using simulated data with zero errors, to determine if the fitting method does actually work for this mode of inhibition and if so, how well.
The presentation of the work is has some other problems. There is potential confusion for the reader in the use of Km, Ki and Vm which should be regarded as constants defining the enzyme. The author frequently refers to the variation of Vm, Km or Ki under varied inhibitor or substrate concentrations– where what is meant is the apparent or measured Vm, Km or Ki. The distinction should be made clear throughout – perhaps by using italicized terms for the experimental values. In places the wording suggests that the Km or Ki represents the affinity of substrate or inhibitor for the enzyme, whereas they are inversely related to the affinity. This should be clarified. Sometimes arguments are presented in prose that is difficult to understand and I recommend a careful reexamination of the text to remove such issues. Instances are given below.
In the spread sheets supporting the paper the results for different types of mechanism are compactly presented in the first sheet, with detailed information about each mechanism in the following sheets. In the first sheet, numerical results are presented, with values for Vms, and various K parameters. These Tables are presented in abbreviated form as Figures in the text. The goodness of fit for each mechanism is illustrated via v vs s and 1/v vs 1/s plots. (I note that in the sheet1 of the S2 file, double reciprocal plots appear to be flat. A scale calculation appears to need adjustment.) Choice of the best fit mechanism(s) is made easily on the basis of RSS values. In some cases the fit to different models is very close. The inclusion of standard errors on the regression parameters would help to distinguish between alternatives in such cases since it is possible that an improved overall fit might result from inclusion of an additional but poorly defined parameter.
This reviewer suggests, perhaps for future adoption, that in the templates for partial inhibition, inclusion of Dixon plots on Sheet 1 , may illustrate the partial nature of the inhibition better than the double reciprocal plots currently shown.
In some cases (e.g. in S9) the values shown in the summary table at the top of sheet are not reflected in the tables for individual mechanisms. Thus in S9 table for the modifier equation, values for Vmax 2 and Ki appear to have been interchanged and Km2 has acquired the value of Vmax.
In other instances, similar confusions may be found. In S13, the modifier equation shown in the bottom box contains the term Kx, rather than Ki. A value for Ki is given in the overall summary, but this value becomes that of Vm in the summary box at the bottom of the sheet.
It is not clear whether the author wishes the global fitting routines to be generally accessible. Pasting new data into the spreadsheets results in no progress beyond the initial estimates. If it is intended that the routines become generally available, some instruction as to the use of the global fitting routine would be valuable.




Minor Issues:
Title: “ Modifier equations” rather than ” modifiers equations”
Line 27: replace “pseudo steady-state” with “Rapid equilibrium”.
Line 30: change sentence after “changes in the…. “ and substitute “measured values of Vmax and Km”
Line 35: change ‘’would be’’ to ‘’is’’.
Line 45: insert ‘’possible’’ after ‘’total” , change “turnover’’ to ‘’conversion’’
Line 77: substitute ‘’absent” for “conspicuously missing”.
Line 85: insert after “plot ” “experimentally determined values of ”
line 90: change “affinity” to “access”
line 92: insert “apparent” before “substrate” and before ”Km”
Line 94: insert “apparent value of” before “Vmax”
Line 98: change “of” to “or”
Line 102 : IC50s
Lines 113-5: Meaning of this sentence unclear.
Lines 116-119: Sentence is incomplete.
For all equations: make sure equation number is aligned with the equation itself.
Line 136 – 139: Meaning is obscure. One assumes that there is a one-to-one association of the inhibitor with the enzyme.
Line 142: processes
LinE 144: “only affect the Km” may read better as “partially increase the measured Km’’
Line 147 – enzyme’s
Line 149: allows
Line 164: Fig 2f. does not illustrate hyperbolic behavior.
Line 167: suggest insertion of “ïn fact’’ after ‘’Km are”
Line 168-169: suggest insert brackets around “as it could………………Kallikrein”
Line 197: Why is a modification of the direct linear plot used? Are there advantages over the original v vs s form that can be described.
Line 224: ‘’much more even”
Lines 276-279: meaning of this sentence unclear –“ particularly strict distribution of lines”.
Line 289: “these, not “theses”
Line 347: “in that it”
Line 348: Equation 7 contains only one inhibition constant. Is 7 the equation that is meant?
Line 362: Intended meaning of this sentence unclear – “access?”

Experimental design

Experimental design:
The work reported here involves the analysis of data obtained from other workers and thus has the merit of objectivity. It is suggested, though, that the use of simulated data with known (or zero) variability would enable a testing of the limitations of the approach.

Validity of the findings

The presentation of this work is a little challenging, as discussed above, but there is no doubt that the analytical approach proposed may simplify and standardize the data analysis of experiments in which the mode of action of inhibitors or activators of enzymes are the focus of the work.

---

## Round 0.2 · Major Revisions

My name is Paul Tulkens and I have been contacted by PeerJ staff to handle the submission as you are appealing the Reject editorial decision made by the prior Academic Editor.

I am issuing this “revised” decision so you can upload the most recent version of your manuscript and a point by point response to the reviewers.

· Appeal

Appeal

Stephen J. Johnson, PhD

Staff Editor, PeerJ

Dear Stephen J. Johnson

Re: Appeal letter against Rejection of my article “Comparing enzyme activity modifier equations through the development of global data fitting template in Excel” for publication in PeerJ

Please consider this correspondence to constitute a formal appeal against the rejection of my article that was notified to me through email February 5 2018.

I would like to challenge the decision to reject the article on the grounds that the editor displayed a clear bias against my work as will be outlined below.

I look forward to your response and hope to resolve this situation promptly.

Yours Sincerely

Ryan Walsh, PhD

During the review process the editor provided a large amount of feedback. Unfortunately, the feedback was biased and overwhelmingly negative, suggesting that I lack an understand the literature, theory and history of the subject matter. This included the analysis of some of my previous publications which were used to try to discredit the article. The outright disagreement with my proposed method for analysing the data resulted in a complete trivialization and disregard of the results which show a clear agreement with the arguments I was making. However, the editor seems to have no interest in allowing the dissemination of my ideas or the templates I have produced which would allow anyone to test the ideas I’m presenting in the article with their own data. He feigned misunderstanding that my belief that inhibition theory was overcomplicated related to the equations themselves being too hard to understand rather than there being too many equations which I suggest can be replaced by one. But I realize such a statement requires a lot of proof which is why the article presents many examples from the literature which are fit to many equations to see how this my equation fairs against other models and an explanation as to how to do the same comparison using any data set. The dissemination of these templates may ultimately prove I’m wrong and if that is the case so be it, but the circular arguments used by the editor to justify his rejection are quite alarming from a peer review. For example, during the initial review he suggested my proposed equation could not describe a plausible physical model as it had not been derived with the classical reaction scheme method, but once this was provided claimed it was incomplete. Additionally, he reversed his belief in the need for physical plausibility by suggesting the classical equations do not require more explanation beyond their initial derivation and do not need to be intuitive. This is an interesting statement to make as I agree things do not need to be intuitive, however the argument I make in the article relates to the inhibition term which I show through rearrangement is a mass action term something that is not apparent by its common or initial derivation in the literature. Once this rearrangement is made it clearly demonstrates that the effect of the inhibitor on the enzyme directly relates to the percent of the enzyme population bound by the inhibitor in an intuitive way. The fact that he is suggesting that this information does not need to be examined is highly unusual.

As my opinion on the situation is obviously biased I have included each of the correspondence from the editor bellow as unaltered text, followed by a response detailing some of my opinions of the points he has raises.

Additionally, I am attaching a recent track changes version of the paper and a response to the last round of reviewer comments

A copy of the updated article and supplementary files can be found at: https://peerj.com/preprints/3094/


· · Academic Editor

Reject

I must admit I am quite disappointed by the limited nature of the changes in your manuscript and by the dismissive tone of several of your responses to the pertinent observations of our reviewers. Specifically:

A) in p. 11 of the response the author states (in reply to reviewer#3) " The advantage of it over the direct linear plot I Believe would be better addressed to the researcher Cornish-Bowden who developed them." This response (as most of the responses to reviewer #2) is unacceptable: since the author decided to use the modified direct linear plot , it behooves him to explain that decision. Referring a helpful reviewer (who took time from their busy schedule to help the author refine his work and make it clearer) to the primary literature instead of clarifying the text for the benefit of your readers does not dispose favorably neither reviewers nor this editor towards your work.

B) As highlighted by reviewer #2, Sebaugh 2011 (which describes guidelines to ensure that sufficient sampling of inhibitor concentrations is performed to obtain a reliable estimate of IC50) is not a proper reference to relate IC50 to inhibitory theory and, while it might be cited, it should not be cited as "distilling inhibitory theory to its most basic form" . The author faults reviewer #2 for suggesting Cortes et al. and Cheng and Prusof as suitable references to the relationship between IC50 and Ki by claiming that the reference to Lazareno and Birdsall, 1993 (which references Cheng and Prusof) as a reference to the use of IC50 is enough. The author fails to note, however, that the specific bone of contention was not the presence of Cheng and Pusof among the reference of a reference, but to the inappropriate citation of Sebaugh as "distilling inhibitory theory to its most basic form" .

C) The derivation of the modified inhibitory equation is incomplete and does not provide more information than in Walsh et al. 2008 and Walsh et al. 2011 (Integrative Biology). This derivation also offers a sui-generis definition of Km as the equilibrium constant of the dissociation of the enzyme-substrate complex, whereas the definition of Km only equals dhe dissociation equilibrium constant when kcat is zero. When the steady-state is achieved, Km does equal the reaction quotient of the dissociation, but that number (although computed in a similar way as the equilibrium constant) is not an equilibrium constant because the whole system is not in equilibrium. As far as I can tell and you have shown, the equation is justified in Walsh et al. 2007 from empirical fit to data, to which it does not fit better than the partial non-competitive inhibition model.

D) in p. 4 of the response, the author states "It can be argued that using 2 dissociation constants to define a single enzyme inhibitor binding event causes confusion". The presence of two dissociation constants is a natural consequence of analyzing two different dissociation events (dissociation of inhibitor from the enzyme:inhibitor complex and dissociation from the enzyme-substrate-inhibitor complex). It is perfectly natural to expect those constants to be different, due to substrate-inhibitor interactions and to substrate-induced/inhibitor-induced changes (subtle or not) on the enzyme surface where the inhibitor (or substrate) binds.

E) in p.6 of the response, the author states "However, the fact that both Baici and Fontes took it upon themselves to redefine the field as recently as 2000 and 2015 more than suggests that there is something wrong with the field [....] " Such an argument proves too much: if atttempts to systematize a wide body of knowledge are taken to mean that "there is something wrong with the field ", then every edition of Chemical Reviews, Annual Review of Biochemistry or a monograph series is an indictment of the current state of research in the respective areas.

F) The author claims in the response that current models of total inhibition suffer from "incomplete derivation of the 1+i/ki term". That term arises in the equations simply from a straightforward (though tedious) application of the steady-state hypothesis to well-defined reaction mechanisms, and I cannot understand why the author regards that specific term and its appearance in the numerator or denominator of different mechanisms as needing a more thorough explanation than that provided by the derivation of the equations, or as indicting the whole field and heralding its demise or irrelevance


Personal review-level comments by the editor:

I understand from other portions of your ouevre that you believe that inhibitory theory is insufficient. Overturning a paradigm, however, requires more than observing a field, seeing it less thriving or less popular than before and using such sociological observations as supporting a claim of a paradigm failure. Convincing current practioners requires showing specific instances where the predictions of the best models of the ruling paradigm fail, showing that common ways of salvaging the paradigm are unsound, and proposing a model which, while preserving all the strengths of the current theory, performs better than the current paradigm. I cannot see how the algebraic reworking of a theoretically-unsupported partial inhibition equation you performed can be seen as an alternative: there is, for example, no systematic way to expand those formulae to enzymes which require two (or more) substrates, or to enzymes where several active sites and allosteric effects are present, unless by using the same steady-state approximations used traditionally. I honestly think that such an opinion is due to an incomplete engagement with the relevant literature and with the way the equations are generated. For example:

1) In the introduction, the author states "“The overcomplication of the enzyme modifier kinetics, epitomized by the definition of 17 types of inhibition and activation interactions (Baici, 2015), seems to be contributing to the demise of the field. " It takes more than relative unpopularity/unsexiness of a field to state that a large number of equations "seems to contribut[e] to [its] demise", or that it has become overcomplicated. As a great scientist once said "Everything should be made as simple as possible, but not simpler" . There are indeed many different inhibitory equations which simply arise from the existence, in several enzymes, of varied intermediate complexes of enzyme and inhibitors, activators, co-substrates, etc. , the possibility of partial inhibition, etc. Such a profusion of mechanisms for different enzymes is not more of "an attempt to salvage a flawed hypothesis" than the postulation of the existence of Neptune by Urbain Le Verrier and John Couch Adams, the transit timing variation method of detecting extra-solar planets, or attributing the observation of periodic stellar distortions to tidal effects of otherwise undetectable planets in their orbit are attempts to salvage "flawed" gravitational or stellar structure theories. I would, in contrast to the author, claim that the possibility of generating such unintuitive (but powerfully explanatory) equations is rather a testament to the power of the algebraic treatment of reaction mechanisms. And I cannot understand how anyone who follows the derivation of an enzyme kinetic mechanism (with the assumptions of fast equilibration of enzyme, substrate, and enzyme:substrate complex and of a steady-state) can claim that modern enzyme kinetic mechanistic theories does not “obey the same universal mass action principle used to define all molecular interactions”.

2) " However this overcomplication may relate to the treatment of Ki in the total inhibitor equations (Eqs 5-8)..In the total inhibitor equations, the Ki is equated to the effect of the inhibitor on the enzymatic activity rather than an equilibrium binding constant marking the concentration where half the enzyme population is bound by the inhibitor.” The author does not articulate how the physical interpretations of mathematical terms could possibly be a cause of "overcomplication": each individual term arising from a King-Altman treatment of a reaction mechanism is composed of a series of individual rate constants which reflect the specific sub-cycle which originates it. There is, in my eyes, no reason to expect all, most, or even any of those terms to have an intuitive interpretation.

3) in 10.7717/peerj.649 "One of the reasons that these models have endured for so long may be related to the false sense of security the equations for these models provide. The common form of the inhibitory term (1+[I]/Ki) found in every equation suggests that it is working in a similar way. This notion is entirely incorrect. The inhibitory term in the competitive inhibition equation (Fig. 1A) directly affects the substrate affinity (K1) by multiplying into it. The inhibitory termin the non-competitive inhibition equation (Fig. 1B) inversely affects the maximum catalytic activity (V1) by dividing into it. A"

As I stated above, once one follows the derivation of the equations, there is no mystery in that term appearing on the numerator (or denominator) - or at least no more mystery than in (e.g.) the different rules used for computing the total resistance of an electric circuit composed of serial versus parallel assemblies of resistances.

4) in 10.7717/peerj.649 "The implications of this are that the non-competitive inhibition equation is the only one of these three equations that is even somewhat correct."

As observed repeatedly in the literature (monographed at least as far back as J. Leyden Webb 1963) the non-competitive inhibition is the only example (among the simple classical cases of uncompetitive, competitive and non-competitive inhibition of single-substrate enzymes) which is almost never found experimentally due to its postulation of no effect of substrate binding on the affinity towards the inhibitor, and should more properly be regarded as a special case of the much more common mixed inhibition model.

Reviewer 1 ·

Basic reporting

no comments

Experimental design

no comments

Validity of the findings

no comments

Additional comments

The methodology was not described as it was suggested. The mathematical methods were not detailed. The methods section remains the same.

Please, explicitily state the objective of this work by describing: "the objective of this work was to..."

Reviewer 2 ·

Basic reporting

.

Experimental design

.

Validity of the findings

.

Additional comments

In my original confidential comment to the editor I said "the author has a lamentably poor knowledge of the literature. The work itself is in general sound, but the views expressed are nowhere close to being original. Numerous previous authors have made the same or similar points". All this remains true, and now I say it directly to the author, as he doesn't seem to have picked up the implications from the comments directed to him.

The revision has been very half-hearted, and the author seems to have been more concerned to justify his own position than to address the reviewers' comments in a serious way. Here I only discuss the responses to my own report, but things could also be said about the other two.

1. "Most new research is a rehash of former work". Yes indeed, and it's particularly true of this paper. That's why there needed to be a proper discussion of previous work.

2. "The reviewer’s inability to provide a reviewed article later than 2001". Has it not occurred to the author that it's HIS job to make a literature study? Reviewers can point at useful directions to take, but he shouldn't expect them to do his work for them. It was not a matter of inability but of disinclination. I can think of at least three books (in addition to Baici's) published in the 21st century that provide some of the review that the author is missing. However, he can look for these himself.

3. No serious reason is given for refusing to cite Cheng and Prusoff. Among the numerous papers that could have been cited, but weren't, there are several from Tipton's group, such as Lizcano et al. (1996) Biochem. Pharmacol. 52, 187-195, that are relevant.

4. "Baici’s non-peer reviewed book". This insulting remark displays an astonishing ignorance of how serious publishers operate. Does the author imagine that Springer publishes books without getting them reviewed? Vanity publishers are, of course, different, but the author would have to be even more ignorant to think that Springer is a vanity publisher. Relatively few books in biochemistry make any mention of peer review (and those that do are usually general textbooks), but that doesn't mean there wasn't any. No, Baici didn't cite Fontes et al., and maybe he should have, but we can only guess at his reason. He also did not refer to Mahler and Cordes, as I would have done if I'd been writing that book, and if I were writing the present paper. Dixon & Webb, too, while we're at it, not to mention J. Leyden Webb (a different Webb). Even Botts and Morales (1954), the first, as far as I know, to discuss the general modifier mechanism.

5. "A valid opinion for biochemists." Yes. In case the author hasn't realized it, enzyme inhibition is a biochemical topic; applications in pharmacology are derivative. To dismiss the opinions of biochemists as of no account hardly deserves a reply.

Reviewer 3 ·

Basic reporting

This paper concerns the creation of an Excel template designed to analyse kinetic data and fit them to a variety of modes of inhibition or activation affecting single-substrate enzymes .The fitting process operates on the basis of minimising the sum of squared residuals and thus permits the selection of the mechanism that gives the optimal fit. This routine will undoubtedly be found useful by many in the field and merits publication. However the presentation of the material still requires considerable revision.
The introduction is long and detailed but lacks clarity. Much is made of the failure by many workers to distinguish between inhibitory effect and the inhibition constant. It is not clear what is meant by the inhibitory effect – is this the relative inhibited velocity (vi /v) or the IC50 or a component of the inhibition equation.? This is not clear and is nowhere defined in the manuscript. As a result , the material presented is difficult to absorb.
The introduction argues, legitimately, that the use of IC50 values to characterise an inhibitor does not define the mechanism. It is also argued that characterising an inhibition mechanism by visual characterisation of Lineweaver-Burk plotting can led to over-simplification. The Excel template offers and objective means of identifying the most plausible mechanism and giving values for the constants defining it. It would be more useful still if calculation of standard errors on the regression parameters were included. Researchers are likely to be more interested in how well individual constants (Km, Vm, Ki) are defined than in the overall goodness of fit.
Several statements in the text refer to values of Km or Vm varying linearly or hyperbolically – but with what independent variable is not stated. It is assumed that the variable is inhibitor concentration. This should be stated frequently enough to remove any confusion. Fig 2 as it stands tends to add confusion to the discussion. All changes in the value of the apparent Vm or Km are represented as linear changes. The figure would be more helpful if they showed, for the different mechanisms, the variation of apparent Km and Vm values with [X] – clearly demonstrating hyperbolic or linear changes.
Since the models employed in the template have been discussed in detail in a previous publication by the author and others, it seems unnecessary to readdress the questions in the detail found here. This section could be clarified by a) showing a reaction scheme embodying all the proposed mechanisms and b) showing the equations generated by the model. without necessarily giving a detailed derivation of each.
In summary, I believe that while the fitting template is potentially of value to workers in the field, the presentation of the work still requires substantial revision.

Experimental design

No comment

Validity of the findings

The author has presented sets of simulated data to demonstrate the merits of the templates. However, the Dixon plots for the data from the partial inhibition simulations are linear. This is not what one expects for partial inhibition (in which the enzyme inhibitor complex retains some activity) . It is also notable that in, for instance, the partial uncompetitive inhibition data set, the fit to the partial inhibition model is the worst of those tried. These data sets need to be carefully checked.

Additional comments

Matters of detail:
The Km does not measure substrate binding affinity – it is inversely related to the affinity. The higher the affinity of the substrate for the enzyme the lower the value of the Km. Similarly, Ki is referred to as a binding constant for the inhibitor whereas it is strictly a dissociation constant.

Line 122: It is implied that partial inhibition mechanisms have been recognised only relatively recently although, in fact, they have been text book material for over 50 years (e.g. Dixon and Webb, Enzymes, 1964).
Line 88. “ubiquitously”
Line 119-122: This sentence suggests that mechanisms or equations are inhibitors.
Line 124 “are believed to affect the activity” would read more accurately as “represent the activity”
185-190: the intended meaning of this passage comes through only with difficulty. In competitive inhibition, the apparent Michaelis constant is given by Km*(1 +[I]/Ki) which is clearly linearly related to the inhibitor concentration.
“In the total inhibitor equations, the Ki is equated to the effect of the inhibitor on the enzymatic activity rather than an equilibrium binding constant marking the concentration where half the enzyme population is bound by the inhibitor. ” Can the author cite instances in which such a confusion is observed?

---

## Round 0.3 · Minor Revisions

I apologize for the long delay it took for me to come with a final opinion on your paper. This was due to combination of heavy current work load, some domestic problems, and the necessity to consult colleagues in order to see whether they could support my opinion on your paper.

I read with attention and interest the paper itself and then looked at the exchange of texts between you and the reviewer who criticized your work.

While at the beginning I was sympathetic to this reviewer and found your way of answering to him/her quite rude, I came to admit that most of your comments in which you dismiss his/her criticisms were largely justified. Also, I found your paper (in its revised version) quite interesting and and worth of publication. Obviously, we may not agree with all what you propose, but it is the author's privilege to present his/her data and conclusions as long as they are not intrinsically flawed and/or contrary to the evidence. In this case, you are presenting an new approach that some of us may find useful (some other may not, however). Thus, my overall assessment is that the paper could be published but that you will need to defend it should the necessity arise.

I, however, would like you to, once again, pay attention to the remarks and comments of the reviewer, and calmly decide whether you could not further modify your text to take some of his/her comments into account. This is the reason why I ticked "minor revision" so that you will have the opportunity to further improve your submission. I did not tick "major revision" because I do not think that the changes needed are really major. However, when you resubmit, ask the Editorial Office to send me your new version (and your rebuttal) so that I may have the opportunity to see exactly how you have finally dealt with the reviewer's comments.

---

## Round 0.4 · accepted · Accept

This is globally an interesting contribution. It raises many questions but give a new look at the topic, which will interest many readers. We may not agree to all that the author proposes, but this is how Science makes progress...

#